# Sleep Quality: A Narrative Review on Nutrition, Stimulants, and Physical Activity as Important Factors

**DOI:** 10.3390/nu14091912

**Published:** 2022-05-02

**Authors:** Monika Sejbuk, Iwona Mirończuk-Chodakowska, Anna Maria Witkowska

**Affiliations:** Department of Food Biotechnology, Medical University of Bialystok, Szpitalna 37, 15-295 Bialystok, Poland; iwona.mironczuk-chodakowska@umb.edu.pl (I.M.-C.); anna.witkowska@umb.edu.pl (A.M.W.)

**Keywords:** sleep, nutrition, physical activity, stimulants

## Abstract

Sleep is a cyclically occurring, transient, and functional state that is controlled primarily by neurobiological processes. Sleep disorders and insomnia are increasingly being diagnosed at all ages. These are risk factors for depression, mental disorders, coronary heart disease, metabolic syndrome, and/or high blood pressure. A number of factors can negatively affect sleep quality, including the use of stimulants, stress, anxiety, and the use of electronic devices before sleep. A growing body of evidence suggests that nutrition, physical activity, and sleep hygiene can significantly affect the quality of sleep. The aim of this review was to discuss the factors that can affect sleep quality, such as nutrition, stimulants, and physical activity.

## 1. Introduction

Sleep is a natural and reversible condition that is controlled primarily by neurobiological processes, and it is a physiological part of human life that is necessary to the maintenance of health and wellbeing [1,2]. Sleep is associated with a reduction in the perception of external stimuli and the cessation of motor activity [1]. The quality of sleep is influenced by many factors, such as diet [3], physical activity [4], and genetic [5] and environmental factors [6]. Sleep has a multifactorial effect on the body: it reduces energy consumption and increases the recovery of the energy storage in the brain, it regulates the adaptive and innate immune response, and it contributes to memory consolidation (the fixing of acquired information in the brain) [2,7,8]. Sleep disorders are associated with the onset and progression of many different diseases, which include cardiovascular disease, depression, and cancer [9,10,11]. Sleep disorders also increase the risk of infectious diseases [12,13,14,15].

In modern times, with a significant increase in the occurrence of both noncommunicable diseases and sleep disorders, our understanding of the factors that are involved in improving the quality of sleep is of great importance. The purpose of this narrative review is to discuss the factors that can affect sleep quality, such as nutrition, stimulants, and physical activity.

### 1.1. Sleep Phases and Duration

Sleep continuity is assessed by the total time of sleep, the delay in falling asleep (i.e., the time between switching off the lights and falling asleep), and by the type and amount of sleep throughout the duration of sleep [14,16]. Physiological sleep consists of two main phases—the REM (rapid eye movements) phase and the NREM (non-REM) phase—which are repeated during sleep. The REM phase is associated with the activation of the sympathetic nervous system, and it leads to an increase in temperature and blood pressure and to an accelerated heart rate [17]. During the REM sleep phase, there is also a decrease in muscle tone [17,18], and activation in the limbic regions, which suggests that REM plays a role in emotional regulation [19]. The NREM phases are longer and are associated with the function of the parasympathetic nervous system and, in contrast to the REM phase, with decreases in body temperature, blood pressure, and pulse. The NREM sleep phase also supports memory consolidation, metabolic regulation, and brain regeneration [17,19].

Sleep disorders that are associated with an insufficient or missing NREM phase are an increasing public health problem that affects the overall functioning of the body [17]. Adults generally spend about 20–25% of their total sleep time in the REM phase, 75–80% in the NREM phase, and they have between four and five NREM cycles [17].

There are large inter- and intraindividual differences in the duration of sleep between people. A study in monozygotic and bizygotic twins indicates the inheritance of sleep duration [19]. The duration of sleep is influenced between 31 and 55%, which shows a significant influence of genetics on the duration of sleep [19]. This study also shows that, not only can the duration be inherited, but also insomnia, habitual sleep time, midday sleep, and the subjective quality of sleep between identical and fraternal twins [19].

In addition to genetic factors, environmental factors, such as the duration and type of work, the distance between home and work, commuting, professional and family responsibilities, and social relationships, also influence sleep needs. In order to enable the body to sleep “healthily”, it is necessary to have adequate sleep duration, regularity, and quality, and the absence of sleep disturbances [20].

The length of sleep of healthy people decreases with aging: a newborn needs 14–17 h of sleep per day, while adults sleep 7–9 h, and older people sleep for 7–8 h [21]. Less than 7 h of sleep is associated with poorer wellbeing and poorer health. In addition, people who sleep less have a higher risk of illness (e.g., depression, mental disorders, coronary heart disease, metabolic syndrome, high blood pressure) when compared with people who sleep a sufficient number of hours (7–8 h) per day [10,12,20].

### 1.2. Insomnia and Its Risk Factors

Insomnia is a clinical condition that is characterized by difficulty in maintaining sleep or falling asleep, and tiredness and irritability during the day [22]. It cannot be determined only by the number of hours of sleep per day. According to the current state of the knowledge, insomnia disorders are found in about 10–20% of the adult population. They are influenced by factors such as a longer duration of sleep, being awake after sleep, respiratory disturbances during sleep, a shortened sleep duration, and sleep fragmentation [22]. Insomnia leads to arousal in the waking state, sleep with an increased metabolic rate, and increased levels of the adrenocorticotropic hormone and cortisol during the early sleep phase [23].

The most common symptoms that are associated with insomnia are sleep disturbance (which occurs in about 50–70% of patients), problems falling asleep, and insufficient regeneration during sleep [23]. Sleep disorders can be classified as secondary (e.g., medicines) or primary (e.g., mental disorders) [23]. Sleep disorders can also manifest as getting up early in the morning, regardless of when one goes to bed, which results in reduced productivity and concentration, irritability, a risk of mistakes and accidents, and lower quality of life [24].

Insomnia in children leads to poor concentration, which can impair learning performance. This may depend on certain situations and behaviors, such as weighing the baby at bedtime or bottle feeding. For example, if there is no stimulus before bedtime, the child may have difficulty falling asleep [24].

Insomnia is a factor that increases the risk of various diseases, such as asthma, gastroesophageal reflux, hypertension, cardiovascular diseases, and type 2 diabetes [24]. The malposition of the hypothalamic pituitary axis in chronic insomnia influences the fluctuations of the thyroid hormones by increasing the concentrations of cortisol, corticotropin-releasing hormone, and thyrotropin [24].

Careful diagnosis is required to detect or rule out insomnia, as there are many sleep-related disorders that patients may mistake for insomnia, such as sleep apnea, restless legs syndrome, or nocturnal cramps [24]. Restless legs syndrome (RLS), which is also known as Willis Ekbom disease, is characterized by uncontrolled movement that is due to discomfort and pain in the legs. The symptoms decrease with movement [25]. Often, other diseases (such as obstructive sleep apnea and nocturnal cramps) are accompanied by insomnia, which is why a thorough diagnosis is so important. The aim of the treatment of insomnia is to improve the quality and quantity of sleep, which thereby improves the wellbeing and quality of life of the patient [23].

In the diagnosis of insomnia, the effectiveness of sleep plays a decisive role (i.e., the ratio between the total amount of sleep and the amount of time spent in bed). Spending too long in bed and in trying to fall asleep is one of the main problems of people who struggle with insomnia. Sleep disorders influence the development of anxiety during and before falling asleep, which also affects the development of insomnia [26]. Cognitive behavioral therapy can be helpful, which aims to treat insomnia by changing thoughts, attitudes, and beliefs about sleep. This therapy aims to reduce negative sleep thoughts, to improve sleep hygiene, and to reduce the hours spent in bed, which result in an improved sleep performance [26].

Every year, more and more people complain about a deterioration in their quality of sleep, and about insomnia, constant sleep problems, waking up at night, and prolonged dreams. The deterioration in the quality of sleep and the incidence of insomnia occur in the general population, but they more often affect women and older people (65 years and over) [12]. There are many factors that influence the development and occurrence of insomnia, such as caffeine abuse, stress at work, the loss of a loved one, divorce, domestic violence, and shiftwork. People who have perfectionism, neuroses, a suppressed personality, or an increased susceptibility to anxiety are more susceptible to sleep disorders [25].

Sleep disorders may also occur in children. They can be influenced, for example, by delayed milestones (in child development, there are certain skills that a child learns at certain points in time), separation anxiety, and hyperactivity [25]. Children’s sleep problems can also be caused by a lack of certain items (such as favorite cuddly toys), stimulation (telling stories before bedtime, reading books, or swinging), and the parents’ absence from the room. These factors can lead to increased anxiety in children, which may cause insomnia [25].

The sleep–wake cycle may be affected by stimulants (such as alcohol, caffeine, and tobacco) and the use of electronic devices. The use of electronics in the bedroom reduces the sleep time and leads to permanent exposure to external stimuli (the sound of a ringing telephone) and a reduction in melatonin release (bright screen light) [27].

### 1.3. Functions of Sleep

Two effector systems are responsible for regulating the immune response (inborn and adaptive): the sympathetic nervous system and the hypothalamic–pituitary–adrenal axis, which are both influenced by sleep. When you sleep too little, the immune system produces a reduced number of antibodies, which are involved in the body’s defensive reactions [28,29].

During sleep, there is a decrease in the release of cortisol, norepinephrine, and adrenaline. The concentration of hormones that affect cell growth, such as growth hormones, melatonin, and prolactin, increases. Prolactin and growth hormone influence the differentiation and formation of new T cells and stimulate the function of type 1 cytokines that control the antigenic response of lymphocytes [29].

Sleep reduces energy consumption (the basic metabolic rate decreases) because, among other things, it reduces the body temperature. The glucose that is consumed by the brain is also reduced: in slow sleep, twice as much glucose is consumed as in the waking state (brain cells have a lower glucose demand). This decrease is not due to an excessively low blood-sugar level, as it is at the same level as when awake [2]. During the REM phase, the metabolic rate increases, which results in increased glucose consumption compared to during the NREM sleep phase [2].

The glymphatic system is a macroscopic system that uses perivascular canal systems to remove certain substances from the central nervous system [30]. The function of the glymphatic system is to remove toxins from the brain that are produced during cellular respiration. During sleep, there is an increased mass flow in which toxins are excreted from the brain [2]. The glymphatic system also contributes to the distribution of glucose, amino acids, lipids, and certain neurotransmitters [29].

As the body becomes older, or if it does not procure enough sleep, the toxin removal may be reduced, which leads to the formation of amyloid plaques, which can be seen in many neurodegenerative diseases, such as Alzheimer’s disease [31].

Sleep and insomnia influence the different connections of the brain. During sleep, there is a spontaneous fusion of the glia and the neurons by the synapses, which leads to the formation of cell networks. The properties of the network are altered by synapses and signal molecules. During sleep, old superfluous memories are erased, new ones are strengthened, and the neuromuscular cycles are strengthened [2].

High blood pressure is a disease of civilization that is the main risk factor for the development of other cardiovascular diseases. High blood pressure is affected by the length of sleep. Studies in humans show that sleep deprivation (≤5 h/day) and insomnia increase the risk of high blood pressure by a factor of five. The risk of high blood pressure is also higher in people who wake up early in the morning (e.g., going to bed late) and who have difficulty maintaining sleep [10,14].

There is an obvious link between insufficient sleep, cardiovascular diseases, and the development of inflammation in the body. The greatest risk of developing cardiovascular disease is in people who sleep for less than 5 h per day. People who sleep for less than 7 h are also at an increased risk of cardiovascular diseases and mortality that are caused by a disturbance of the functioning of this system. The duration of sleep should, thus, be at least 7 h [13,14]. 

Insufficient sleep leads to increased concentrations of inflammatory markers, such as C-reactive protein (CRP), while inflammation is thought to be related to the incidence of breast cancer and lung tumors [14]. People who work long hours on night shifts have an increased risk of cancers, such as breast cancer, colon cancer, and non-Hodgkin’s lymphoma [32].

Sleep disorders and insomnia may occur as one of the symptoms during depression. Studies show a doubly increased risk of depression in people with sleep disorders [33]. Increased markers of inflammation, which can be caused by insufficient sleep time, are often high in people who suffer from depression. Experimental studies in which inflammation has been activated in the body have shown that the symptoms of depression are accompanied by the increased activation of the brain, and especially the areas that are responsible for regulating the negative and positive effects [9,11,14,33].

## 2. Search Strategy

In order to present the scientific reports of the last decade (2012–March 2022), the electronic databases Medline and the Web of Science were investigated. Publications on sleep quality in relation to nutrition, stimulants, and physical activity in observational studies, experimental studies, and meta-analyses were collected. This search included human studies and animal experiments. In some cases, studies that are important for this review but that are outside of the search period are described. The search terms were multiple, with “sleep” as the main search term, in combination with “insomnia”, “nutrition”, “protein”, “carbohydrate”, “fat”, “vitamins”, “vitamin D”, “minerals”, “tryptophan”, “melatonin”, “gamma amino butyric acid”, “GABA”, “caffeine”, “nicotine”, “alcohol”, “marijuana”, “cannabis”, and “physical activity” as the most critical for this review. The full-text articles published in English that were relevant for this review were selected. To qualify the publications for further evaluation, the titles and abstracts were initially screened according to the search criteria. Studies that did not meet the search criteria were excluded.

## 3. Review

### 3.1. Diet and Sleep Quality

Proper nutrition involves providing all of the necessary nutrients in order to maintain health and wellbeing. The foods that people consume can not only influence their wakefulness during the day, but also their quality of sleep. Sleep is not only influenced by the energy efficiency of the diet, but also by the content of macronutrients, such as proteins, carbohydrates, and fats [34]. Insufficient protein intake may impair sleep quality, while too much protein intake may lead to difficulties in maintaining sleep [34,35].

An important relationship has been identified between the quality of carbohydrates ingested (fiber content of the products and the degree of processing of the food) and the quality of sleep [35]. The glycemic index and the frequency and time of meals are influenced not only by the carbohydrate intake, but also by the quality of the carbohydrates that are consumed [35]. The high consumption of noodles, sweets, and sugary drinks, as well as the omission of breakfast and irregular meals, are associated with poor sleep, while a diet that is rich in fish, seafood, and vegetables contributes to good sleep. The inadequate intake of macronutrients, excessive calorie intake, and late meals contribute to a reduction in sleep quality and may influence the development of insomnia [35]. 

Eating foods that are rich in tryptophan, melatonin, and serotonin improves sleep quality. In adults, after consuming foods rich in tryptophan, a longer downtime, increased performance, and an increased total sleep time have been observed [36]. Vitamins and minerals (e.g., B vitamins, zinc) influence sleep quality, and when a deficiency was compensated for, an improvement in the sleep rate and the overall sleep quality was observed [37].

Increasingly, a tendency to sleep for less than 6 h per night is being observed in the general population, which is influencing the increased consumption of coffee, which contains caffeine. The half-life of caffeine averages 2 to 10 h, but it can be up to 20 h. Caffeine obviously increases performance, but it also has side effects: it affects the quality of sleep. People who consume large amounts of caffeine are more likely to be drowsy in the morning than those who consume moderate amounts [37,38].

Adults often drink alcohol, and some of them believe that alcohol even helps them to fall asleep. However, alcohol has a negative effect on sleep, and it impairs the electrophysiological structure of sleep, affects biorhythms, and increases insomnia. In studies that used moderate doses of alcohol (<1 g/kg body weight), there was a shortening of the REM sleep phase, which was mainly in the second half of sleep [39]. 

Nutrition also has a significant influence on the wellbeing of sleep. However, the nutritional mechanisms that influence sleep regulation are complex [35]. Individual ingredients in the diet can directly influence sleep, such as caffeine, which prolongs the duration of sleep induction but reduces the overall duration and quality of sleep [40]. Many food metabolites may be important in the regulation of sleep through the regulation of other related factors. Foods may also influence the commensal microbiota, which may lead to the formation of metabolites [41]. Inadequate nutrition in the long term may contribute to inflammation, which is closely related to insomnia [42]. Adequate nutrition that is rich in fruits, vegetables, and whole grains has a positive effect on sleep [43].

#### 3.1.1. Sleep and Energy Intake

Overweight and obesity are a growing problem in developed and developing countries. Overweight and obesity influence many concomitant diseases, such as type II diabetes, cancer, and cardiovascular diseases. Inadequate sleep habits and poor sleep hygiene may correlate with overweight and obesity [44]. The available data show that up to 95% of pupils in upper secondary education do not meet the requirements for adequate sleep time [45]. In the last hundred years, sleep duration has decreased by one hour in all age groups [46]. The available data also show a decrease in the sleep duration in recent years of 10–15 min, and an increase in the number of people who sleep less than 6 h [47].

Short sleep has been shown to increase the risk of developing obesity. Insufficient sleep leads to increased food intake, which leads to an excessive calorie diet. Studies have shown a link between insufficient sleep duration and biological changes in hunger [44].

Insufficient sleep is associated with hormonal changes in the body, including the release of leptin, ghrelin, cortisol, and growth hormone. Hormonal changes may result in reduced tissue insulin sensitivity. These changes have an impact on inappropriate food selection, changes in energy regulation, excessive food intake, and reduced physical activity [48,49].

People who sleep less have a shorter REM period, which probably plays a role in the link between weight gain and insufficient sleep. A study with 335 participants showed significant differences between the sleep phases in overweight children and those of children of adequate body weight [50]. The subjects with excessive body weight had reduced sleep function, a longer delay in the first REM phase, reduced REM time, and reduced REM activity.

Leptin and ghrelin are hormones that are involved in the regulation of appetite. Ghrelin is responsible for the feeling of hunger, while leptin is responsible for the feeling of satiety [51]. The level of leptin in persons with insufficient sleep durations decreases, while the level of ghrelin, which leads to a subjective feeling of hunger, is increased [52,53]. Poor sleep hygiene, which goes hand in hand with insufficient sleep time, has an impact on poor dietary choices, such as increased portions, increased calorie intake, an increased feeling of hunger, and increased consumption of sugary drinks and foods [49].

Our current knowledge shows a connection between sleep quality and obesity. Overweight and obese individuals have been shown to have lower quality of sleep than those of normal body weight, regardless of the length of sleep [49,54], which involves the number of wakes within 5 min, the sleep performance, delayed sleep, and waking up after sleep.

High energy and fat consumption, binge eating, and nighttime snacking lead to sleep disorders, which can subsequently lead to disturbances in the feelings of satiety and hunger. Short sleepers prefer high-calorie foods and frequent snacks, and they skip meals more frequently [51]. 

The relationship between poor sleep quality and the development of overweight/obesity is illustrated in Figure 1.

Sleep disorders have a significant impact on people’s quality of life. Proper nutrition can significantly improve the quality of sleep. A balanced diet should contain all of the necessary minerals, vitamins, and amino acids. Poor nutrition can influence the onset of insomnia, which, in turn, is a factor in the development of many serious diseases, such as high blood pressure, type 2 diabetes, and cardiovascular diseases. Food-borne substances may impair sleep quality (e.g., by causing inflammation or alterations in hormone regulation) [41].

#### 3.1.2. Dietary Fat and Sleep Quality

Nuts, vegetable oils, and olive oil are characterized by high contents of unsaturated fatty acids, but low contents of saturated fatty acids. The consumption of these products is lower than recommended for the majority of the population that is in favor of saturated fats. The excessive consumption of foods that are rich in saturated fatty acids contributes to the development of noncommunicable diseases (NCDs) [55,56,57]. Studies also show that people with insomnia have a higher consumption of high-fat foods than people without sleep disorders [35].

Eating fatty fish (more than 5% fat, such as salmon, mackerel, and trout) has a positive effect on sleep regulation. Fatty fish are a good source of omega-3 and omega-6 fatty acids, as well as of vitamin D. These nutrients may influence the regulation of serotonin secretion and, thus, the regulation of sleep [58]. Eating fatty fish leads to an increased feeling of drowsiness, which leads to better sleep and a more efficient performance during the day. Current evidence suggests that the consumption of fatty fish may have a positive impact on daily functioning and sleep [58].

Polyunsaturated omega-3 fatty acids are an important component of the diet. Diets that are low in omega-3 acids may impair sleep at night because of an endogenous disturbance of the daily clock and a reduction in melatonin secretion. Studies in hamsters with omega-3 deficiency have shown a disturbance in the melatonin-secretion rhythm and chronic locomotor hyperactivity [59,60].

Animal fats contain, almost exclusively, saturated fatty acids. Foods that are fried in hydrogenated oil are also a rich source of saturated fatty acids [59]. Studies on the effect of saturated fatty acids on sleep have shown that the consumption of saturated fatty acids leads to a greater number of wakes at night and shortens the duration of slow-wave sleep, which is the stage of sleep during which the body can recover [59]. The regular consumption of saturated fatty acids contributes to the development of diabetes, which is often associated with sleep problems [61].

The effect of dietary fat on sleep quality is shown in Figure 2.

Of the unsaturated fatty acids, arachidonic acid, which is a precursor to the production of the prostaglandin PGD2, which is a sleep-promoting eicosanoid, is of crucial importance for sleep quality [62].

#### 3.1.3. Dietary Protein and Sleep Quality

Protein is one of the three main nutrients that meets the body’s energy needs. Between 1999 and 2016, there was an increase in the estimated consumption of protein-source products, which was associated with an increased consumption of poultry, eggs, and soy [63]. Protein requirements change with age and are dependent on the condition of the body. For example, people who are ill or who have extensive burns will have an increased need for protein. Proteins have many functions in the body, such as transport, building, and structural functions [64].

A study of 4435 nonshift workers showed that protein intake can influence the symptoms of insomnia [65]. Low protein intake (<16% of total energy) was associated with a difficulty falling asleep and poor sleep quality, whereas high protein intake (≥19% of total energy) was linked to difficulty maintaining sleep. On this basis, it is recommended that protein accounts for between 16 and 19% of the energy efficiency of one’s food intake [65].

Protein may consist of the amino acid tryptophan, which is a precursor of cerebral serotonin, which acts as a sleeping pill. Too little protein intake can lead to a deficiency of tryptophan, which can lead to sleep disturbances [66]. However, an excess of protein in the diet may lead to a decrease in the level of tryptophan in the brain, since the protein also contains other large neutral amino acids (LNAAs) (wide neutral amino acids) that affect the transport of tryptophan by the blood–brain barrier of tryptophan [66].

Protein that is taken in the evening has a positive effect on muscle protein synthesis during sleep. The availability of amino acids overnight is limited, and the rate of muscle protein synthesis is limited; however, by absorbing protein before bedtime, it is possible to digest and absorb it effectively. During sustained resistance training, the consumption of protein before bedtime may additionally influence muscle buildup and muscle strength [67].

The effect of proteins on sleep quality is presented in Figure 3.

#### 3.1.4. Dietary Carbohydrates and Sleep Quality

Dietary carbohydrates, and the degree of their processing, significantly affect the quality of sleep. Both low-carbohydrate and high-carbohydrate diets affect the sleep architecture [35]. Carbohydrates have been shown to affect mainly the NREM phase (slow-wave sleep) and the REM phase. Moreover, dietary carbohydrates may also delay the onset of the REM phase and may delay the onset of sleep [34,68].

The carbohydrate quality is even more important for sleep quality than the amount of dietary carbohydrates. A study that was conducted in a group of 12 healthy subjects who were aged 18–35 years, and who consumed a meal that contained carbohydrates with a high glycemic index (GI) four hours before bedtime, showed a significant reduction in the delay in falling asleep, compared to a meal that contained low GI products [69].

Other studies suggest that a diet with a high glycemic index is a factor that increases the risk of insomnia [70]. The “Women’s Health Initiative Observational Study”, which was conducted with the participation of postmenopausal women, examined the probability of insomnia after consuming carbohydrates with different glycemic indexes, glycemic loads, and fiber contents. The risk of insomnia was increased by products with higher glycemic indexes and higher amounts of added sugars, refined grains, and starches. By contrast, a higher consumption of dietary fiber, whole grains, fruit, and vegetables was associated with a lower risk of insomnia [70]. Higher dietary fiber content in food lowers the glycemic index and slows down the metabolism of carbohydrates [71].

The mechanism of carbohydrate consumption on the occurrence of insomnia is not yet fully understood, but potential mechanisms have been suggested. By consuming foods with high glycemic indexes, the concentration of insulin increases, which may change the ratio of tryptophan to other large neutral amino acids (LNAAs), such as leucine, isoleucine, phenylalanine, valine, methionine, and tyrosine [72]. Insulin affects the higher selective uptake of LNAAs by the muscles, which thus causes a higher ratio of tryptophan compared to these amino acids. LNAAs compete with tryptophan for transport to the brain, and the greater muscle uptake of amino acids may lead to increased levels of tryptophan in the brain [73]. Tryptophan, on the other hand, is a precursor of serotonin, which affects sleep; therefore, the consumption of a large amount of carbohydrates with a high glycemic index may improve sleep wellbeing. However, in a proper diet, carbohydrates with a high glycemic index should be avoided because they contribute to the development of NCD, such as diabetes type 2 [69]. For a meal to have such an effect on the body, it should contain only carbohydrates. Even if only 5% of a meal comprises protein, it may inhibit the increase in the tryptophan concentration in the brain [70].

A diet with a high glycemic index may cause hyperglycemia, and the resulting hyperinsulinemia may induce the release of hormones such as cortisol, growth hormone, glucagon, and insulin, which contribute to sleep disorders [71]. A diet with a high glycemic index may deteriorate the quality of sleep by stimulating the inflammatory immune response, which leads to changes in the intestinal microbiome [34].

Some studies have examined the influence of diet on sleep quality. In a group of 26 adults, who usually slept for 7–9 h per day, the consumption of products that contained little dietary fiber and high amounts of saturated fatty acids resulted in less deep sleep [74]. The consumption of refined carbohydrates and sugar resulted in an increased number of awakenings during sleep [74].

Sleep quality is also influenced by the relationship between the percentage of energy that is consumed from sugar and nonfiber carbohydrates during the day. An increased probability of the reduced regularity of sleep and wakefulness has been shown in people who consume higher amounts of carbohydrates (i.e., ≥70.7% of energy comes from carbohydrates), compared to people who consume a moderate amount of carbohydrates (61 to 66% of energy) [75].

A meal that is abundant in carbohydrates and that is eaten in the evening reduces the nocturnal secretion of melatonin and delays the circadian rhythm of the basal body temperature [75]. The consumption of fiber was associated with more regenerative and deeper sleep. It is possible that a diet that is high in crude carbohydrates, contains more fiber, and with a reduced consumption of nonfibrous carbohydrates and sugar may significantly improve the quality of sleep of people with insomnia [34]. A study of 410 young women found that those who slept less consumed more carbohydrates and less dietary fiber [76]. 

The effects of carbohydrates on sleep quality are shown in Figure 4.

#### 3.1.5. Caffeine

One of the most commonly consumed stimulants is caffeine, which is found in coffee, tea, chocolate, energy drinks, and carbonated drinks. These products are widely used and are also consumed by children and adolescents [34].

Caffeine is, among other things, a substance that is taken to relieve fatigue; however, abuse can have a negative effect on sleep [37,77,78]. Therefore, it is very important to eat an adequate and balanced diet in order to enjoy complete mental and physical health [40,41].

The half-life of caffeine is between 2 and 10 h, depending on exogenous and endogenous factors. Nicotine can change (accelerate) the rate of caffeine metabolism by up to 50%. The residual effect of caffeine may last longer than 10 h, or even up to 20 h [37].

The mechanism of action of caffeine on the central nervous system is adenosine-receptor antagonism. Consequently, the effect of caffeine on sleep quality appears to be mainly due to adenosine receptors [38,79]. Most adults consume daily caffeine (coffee, energy drinks, tea, or other drinks), with an average intake of 200 mg/day. Self-reported studies tend to underestimate the actual caffeine consumption. This is because many people miscalculate the caffeine content of their diets by stating only the amount of caffeine in the coffee that they consume, without indicating the caffeine content in cold medicines, painkillers, tea, chocolate, hot chocolate, and energy drinks. It is difficult to develop a completely decaf diet, considering how widespread and easily accessible caffeine is [37].

Caffeine acts, in particular, on the A1 and A2A receptors, which influence sleep, cognitive functions, and the arousal of the brain. The absorption of caffeine takes place in the small intestine and stomach. This process is fast and effective, with the maximum plasma concentration being reached within the first 30 min after caffeine absorption [37,38].

For most people in the world, sleep is the time when they stop consuming caffeine. The negative effects of caffeine consumption (too much or too late) can only be felt the next day [37]. The intake of caffeine-containing coffee results in a decreased secretion of 6-sulfatoxymelatonin, which is the main metabolite of melatonin [77]. This is one of the mechanisms that causes sleep interruption [37].

The administration of four cups of brewed coffee (equivalent to 400 mg of caffeine) up to 6 h before bedtime leads to a significant deterioration in sleep quality. Caffeine consumption, even in the morning, shifts the REM phase of sleep to the early night [37]. 

In one study, nine healthy male volunteers were administered 200 mg of caffeine in the morning (at 7:00) [80]. The sleep episodes were monitored by electroencephalography, while the caffeine concentrations were measured in the saliva. A sharp increase in the caffeine levels was observed up to one hour after taking 200 mg of caffeine. A decrease in the caffeine concentration to less than one fifth of the peak level occurred 16 h later. Despite the decrease in the amount of caffeine in the saliva at the time of falling asleep, the overall time and efficiency of the sleep was reduced. This study shows that even a moderate dose of caffeine taken in the morning negatively affects the sleep quality during the subsequent night.

A study of 309 children aged 8 to 12 years investigated the relationship between the sleep quality, caffeine consumption, and daytime behavior [81]. It looked at caffeine intakes of between 0 and 151 mg (with the latter corresponding to an intake of 500 mL) of the energy drink Red Bull, or 750 g of milk chocolate. The intake of caffeine affected the sleep quality, the morning fatigue, and the sleep routine. The most common sources of caffeine among the children were coffee and tea (41%) and carbonated beverages (40%). No effect of caffeine on sleep delay in children has been observed.

Caffeine consumption causes an increased number of naps, a shortened total sleep time, a poor subjective assessment of the sleep quality, and daytime sleepiness [80].

#### 3.1.6. Vitamin D

Vitamin D is a fat-soluble vitamin that plays a crucial role in calcium absorption. The main source of vitamin D is skin synthesis (ultraviolet B). It can also be supplied through food (fatty fish is the main source). On the basis of the serum concentration of the major active metabolite of vitamin D (25-hydroxyvitamin), vitamin D deficiency is widespread [82,83,84].

Low serum levels of 25-hydroxyvitamin D may impair sleep quality. The relationship between sleep disorders and vitamin D deficiency is not fully understood [85].

Vitamin D receptors are widespread in almost all tissues, including in the central nervous system [86]. Vitamin D receptors are found in the human brain in the prefrontal cortex, hypothalamus, and in black or grey brain matter, all of which play an important role in the regulation of sleep [87]. 

Vitamin D deficiency can cause nonspecific pain, which can impair sleep and worsen sleep quality. People who complained of nonspecific pain of an unknown cause had an increased risk of shortened sleep duration and worsening sleep quality. A study of 28 US veterans with vitamin D deficiencies showed that vitamin D supplementation can improve sleep quality and duration, relieve pain, and improve quality of life [88].

Vitamin D deficiency is associated with a higher risk of insomnia, including short sleep duration, poor quality of sleep, and daytime sleepiness. Studies suggest a correlation between a deterioration in the sleep quality and a deficiency of 25-hydroxyvitamin D in serum [89].

#### 3.1.7. Tryptophan, Serotonin, and Melatonin

Tryptophan is an essential amino acid that the body cannot synthesize itself and, thus, must be supplied through food. Sources of tryptophan include chicken, turkey, eggs, milk, fish, cheese, beans, and pumpkin seeds [90]. Tryptophan is a precursor of melatonin and serotonin that can cross the blood–brain barrier and compete with other major neutral amino acids [91,92]. The conversion of tryptophan into serotonin occurs under conditions in which tryptophan is sufficiently available in the brain. Increased levels of tryptophan in the brain occur when the ratio of free tryptophan to branched-chain amino acids is increased. Melatonin is formed in the process of tryptophan conversion into serotonin [92,93].

Relatively low doses of tryptophan in the diet increase sleep performance, shorten the waking time during the night, and increase the subjective assessment of the sleep quality [92]. A study was conducted with 35 subjects aged 55 to 75 years who had trouble sleeping. The participants received a meal of 30 g of flakes with 22.5 mg of tryptophan in the first week, 30 g of flakes with 60 mg tryptophan in the second week, and a normal meal in the third week. The study showed a significant improvement in the sleep quality in the second week with a flake diet of 60 mg tryptophan, compared to the first and third weeks. An improvement in the quality of sleep was particularly evident in the performance of sleep, the increase in the actual duration of sleep, and the time during which one was immobile [92].

The soothing hormone that is released by the pineal gland is melatonin. Increased exogenous melatonin levels can improve sleep quality by increasing the body temperature [93]. Melatonin is mainly produced in the dark by the pineal gland from serotonin, and it stimulates the circadian rhythm. With increasing age, the melatonin level decreases, which leads to a disturbance in the circadian sleep rhythm [92]. Melatonin can be supplemented. Melatonin from supplements is characterized by very low toxicity, although no additional benefits have been observed at doses above 3 mg. The absorption of melatonin has a positive effect on the quality of sleep by increasing the propensity to sleep and by increasing the duration of sleep [42].

#### 3.1.8. Gamma Aminobutyric Acid

Gamma-aminobutyric acid (GABA) is a nonprotein amino acid that has a positive effect on many metabolic disorders. The main producers of gamma-aminobutyric acid are lactic acid bacteria [94]. High concentrations of GABA in food can be achieved by the use of *Lactobacillus brevis* or *Lactococcus lactis*, which are present in fermented dairy products. GABA occurs naturally in small amounts in rice, oat, wheat, soya beans, raw spinach, potatoes, and many vegetables [95,96]. Naturally occurring gamma-aminobutyric acid can also increase sleep efficiency and can have hypnotic effects [97].

One study investigated the objective effect of the uptake of naturally occurring gamma-aminobutyric acid by means of serial polysomnography. The study included adults with one or more symptoms of insomnia. The patients received gamma aminobutyric acid tablets or a placebo one hour before bedtime. The patients who received gamma aminobutyric acid tablets had a significantly reduced sleep delay compared to the patients who took placebo tablets. A reduction in the symptoms associated with insomnia and a subjective improvement in the quality of sleep has also been observed in patients taking GABA tablets [97]. Although the study did not investigate whether GABA can cross the blood–brain barrier, it is suspected that gamma aminobutyric acid may inhibit stimulation neurons [97,98].

Gamma aminobutyric acid tablets can affect sleep by enhancing central GABA-ergic neurotransmission. Gamma-aminobutyric acid is involved in the regulation of both the REM and the NREM sleep phases [96,99].

### 3.2. Stimulants and Drugs That Affect Sleep Quality

Sleep hygiene is a set of environmental and behavioral recommendations that are aimed at the promotion of healthy sleep. It should be used in the entire population, and not only in persons with insomnia [100]. Patients are instructed to adhere to the rules of proper sleep hygiene (stopping smoking, avoiding alcohol, regular sleeping hours, regular exercise, and noise avoidance). There are people who do not have access to sleep therapy, even though they meet the criteria for sleep disorders. Such people will more often look for materials to help deal with insomnia themselves and turn to basic care officials. Information about proper sleep hygiene is easily accessible and widespread, as it can be disseminated without the direct involvement of a doctor. As a result, it can also be accessible to people who are not seeking medical help for sleep disorders [101]. 

Education about proper sleep hygiene is relatively inexpensive and may be the first intervention for people looking to improve their sleep quality. The most common sleep hygiene recommendations refer to smoking, alcohol, caffeine, daytime naps, stress, noise, time outdoors, and exercise [102]. It is important to avoid the factors that influence the deterioration of sleep quality, such as smoking, alcohol consumption, excessive stress, or excessive caffeine intake in the diet [103].

#### 3.2.1. Alcohol

Alcohol is not recommended before going to sleep in order to ensure good sleep hygiene. Alcohol consumption in the late evening prolongs slow-wave sleep in the early part of the night [103] and affects the REM phase and sleep continuity. Alcohol is often used as a trigger for sleep, although the mechanism for reducing sleep delay is more complicated. High doses of alcohol (1 g/kg body weight) within one hour before bedtime inhibits the REM phase, but the reduction in the REM phase disappears with continued alcohol consumption. With high and moderate alcohol consumption, slow-wave sleep is prolonged. Repeated nighttime alcohol consumption leads to a decrease in the NREM phase (slow-wave sleep). Alcohol has a stimulating effect when consumed in low doses (0.16 g/kg body weight) in the first hour after consumption, while it has a sedative effect in large quantities. Alcohol consumption up to 6 h before bedtime impairs the quality of sleep, which indicates a relatively long effect [104].

Increasing alcohol consumption decreases the delay in falling asleep. Two to three hours after drinking, the blood alcohol level drops, which increases arousal. There is a prolongation of the REM phase in the second half of the night, which contributes to the fragmentation of sleep [104]. In long-term users, only a small improvement in sleep can be observed after prolonged abstinence [105,106,107]. Long-term alcohol consumption may have different effects compared to people who drink little or no alcohol. Alcohol abuse can lead to changes in physiological sleep and waking, which play a role in the regulation of sleep [104,106].

Alcohol is considered a good sleeping remedy, not least because of its easy availability and low cost. Women use alcohol less frequently than men [105]. A total of 67% of patients who complained of insomnia (age range: 18–79 years) and used alcohol reported that it had a positive effect on their sleep quality [108]. However, alcohol quickly loses its sleep-promoting effect and retains its sleep-disrupting properties. Alcohol users are also sleepier during the day than nonalcohol users [105].

Low alcohol consumption can cause snoring and obstructive sleep apnea in healthy people. Alcohol in combination with obstructive sleep apnea increases the risk of strokes, heart attacks, and sudden death. The onset of sleep apnea deteriorates the quality of sleep and causes the feeling of fatigue during the next day [105,109].

Alcohol consumption aggravates movement disorders, which impairs sleep behavior. People who consume two or more drinks a day have a two to threefold increase in periodic leg movements, which leads to the increased fragmentation of sleep [108].

Alcohol consumption can also cause other sleep-disturbing symptoms, such as gastritis, gastroesophageal reflux, and polyuria. Increased thirst and polyuria cause frequent awakening, which also affects sleep quality [105]. 

#### 3.2.2. Nicotine

Smokers have a higher risk of sleep disorders such as sleep apnea, sleep disturbances, poor sleep quality (increased sleep delay, shorter time, and greater difficulty in maintaining sleep, as well as daytime sleepiness), and insomnia [110].

Nicotine disturbs the balance of neurotransmitters that are involved in the regulation of sleep. In addition, nicotine withdrawal occurs during sleep, which affects the onset of insomnia. In exploratory studies, a significant interaction was observed between evening nicotine intake and the reported occurrence of insomnia. In individuals with symptoms of insomnia, nicotine intake at bedtime has also been associated with a 40 min reduction in sleep duration [110].

Nicotine promotes excitement and alertness by stimulating the cholinergic neurons in the basal region of the forebrain. The intake of nicotine in the form of a patch, a pill, or smoking is associated with sleep disturbances. The administration of nicotine in any form reduces the total sleep time, increases sleep delay, suppresses slow-wave and REM sleep, and increases early morning awakening. Therefore, it is recommended to avoid nicotine in order to maintain good quality sleep [111]. 

People who use nicotine and are addicted to it should be individually investigated for the direct effects of nicotine withdrawal on sleep quality. The withdrawal of nicotine in the early stages of smoking cessation is often associated with the onset of sleep disorders. A deterioration in sleep quality may occur up to 3–4 weeks after quitting smoking. The most noticeable discomfort during nicotine depletion is the more frequent and prolonged awakening during sleep [110,112].

#### 3.2.3. Cannabis

Cannabis (also known as marijuana) is one of the most commonly used drugs. Cannabis has a positive effect on sleep quality. It is sleep-promoting and hypnotic, but it also reduces the waking time after falling asleep, reduces the delay in falling asleep, shortens the REM phase, and prolongs the duration of slow-wave sleep [113,114,115,116].

A 2017 New England study on 1500 patients found a reduction in sleeping-pill use of two-thirds among patients using medicinal marijuana [117]. The hypnotic effects of cannabis are often the reason for its use in people with sleep disorders. However, in cannabis users, the hypnotic effect may be tolerated because of neurological changes in the endocannabinoid system [118].

Cannabis, and especially if used for a short time, can have a soothing effect on sleep disorders in terms of subjective sensations. With the prolonged use of cannabis, however, negative effects on sleep quality have been noted, and most notably during withdrawal. The negative effects of cannabis use can also be observed in individuals who take low doses of cannabis [119,120].

People who use cannabis on a daily basis are more likely to report sleep disturbances compared to people who use cannabis rarely or never [121].

The medical use of cannabis for the treatment of post-traumatic stress disorder, pain, and multiple sclerosis can improve sleep quality. However, long-term cannabis use causes tolerance, long-term sleep disorders, and withdrawal symptoms, which may lead to the worsening of post-traumatic stress disorder [118].

A study was conducted with eight subjects who orally received a fluid that contained naturally occurring substances in marijuana. The volunteers received a polysomnography to assess the sleep and morning functions. Decreased performance on the following day (mood swings and memory deficits) and a deterioration in the sleep quality (decreased sleep performance) were observed [118].

In a sleep study that used polysomnography, it was shown that the administration of 10, 20, or 30 mg of one of the substances contained in marijuana (THC) resulted in a shortening of the latency of falling asleep, and a reduction in the total time of falling asleep. However, not all studies show this effect of marijuana, which may be due to the soporific effects of THC and the stimulant effects of cannabidiol, which is one of the active chemicals that is identified in cannabis [119]. Some studies have also shown that cannabis shortens the rapid-eye sleep phase, reduces the density of the rapid-eye sleep phase, and lengthens the NREM (slow-wave sleep) phase. Long-term marijuana users develop a tolerance to the effects of cannabis, but they also experience a deterioration in sleep performance [119].

Numerous studies on marijuana withdrawal have shown an increase in wakefulness after falling asleep, an increase in sleep latency with rapid eye movement, an increase in the delay of falling asleep, and decreases in slow-wave sleep, sleep performance, and total sleep duration. Such effects are more pronounced in people who use marijuana intensively (marijuana use ≥5 days a week for the last 3 months). Symptoms can persist for more than 45 days [119].

More and more people are complaining about a decrease in the quality of sleep or the occurrence of insomnia. Proper nutrition that is rich in tryptophan, vitamin D, and gamma-aminobutyric acid can improve the quality of sleep. By using foods that are rich in these substances, the effectiveness and the actual sleep time are improved. In addition, there is a noticeable delay in the subjective assessment of sleep.

Substances such as alcohol, nicotine, excess caffeine, and cannabis negatively affect the quality of sleep. They cause, among other things, an increase in the waking time after falling asleep, a shorter sleeping time, and greater difficulty in maintaining sleep. A nonpharmacological method of treating insomnia is to eliminate the consumption of the abovementioned substances [87,104,110,121]. The effects of caffeine, alcohol, nicotine, and marijuana on sleep quality are shown in Figure 5.

### 3.3. The Impact of Physical Activity on the Quality of Sleep

Sleep and physical activity are related to cognitive functions, and especially executive control and memory consolidation. However, it has not been discovered how physical activity is related to the executive-control processes that are responsible for monitoring, initiating, and planning target-oriented behaviors (e.g., working memory) [122,123]. The quality of sleep in the elderly is affected by physical activity. Longer daily activity is associated with better sleep quality. A study was conducted in which three age groups were included: young (21–29 years), middle-aged (36–64 years), and older (65–81 years). The older people who engaged in more physical activity scored lower on the PSQI (Pittsburgh Sleep Quality Index), which indicates better sleep quality. The older people who slept better reported less fatigue. A relationship between sleep quality and training intensity has also been observed. Moderate and intense physical activity has a positive effect on sleep quality, while light physical activity has no effect on sleep quality [124]. People who practice physical activity sleep better and longer than those with a sedentary lifestyle. By introducing the appropriate amount of physical activity and the time spent outdoors, and by engaging in activities such as walking, we can nonpharmacologically improve the quality of sleep [125]. Long-term physical activity has a positive effect on the quality of sleep. An improvement in the sleep quality occurs with an increase in the activity time and the number of steps, and so even moderate physical activity has a positive effect [126]. High-intensity exercise during the nighttime period affects the secretion of melatonin and can quickly change its concentration in the body within a few minutes [126]. The concentration of melatonin depends on the intensity, duration, and type of exercise that is performed. Physical exercise late in the evening, when melatonin is physiologically secreted, can cause a decrease in its concentration. On the other hand, night exercise, both of high and moderate intensities, causes a delay in the secretion of melatonin on the following evening. Physical exercise during the day, regardless of intensity, does not have a quick and constant effect on melatonin secretion [127].

The effects of physical activity on sleep quality are shown in Figure 6.

#### 3.3.1. The Effect of Recreational Physical Activity on Sleep Quality

Physical activity, physical fitness, and exercise are interrelated, but are, at the same time, separate collective terms. Physical activity is any movement that causes energy expenditure, and it includes, among other things, daily duties, such as household activities and commuting. Exercise is a structured and repetitive activity that is aimed at improving health or at maintaining it at a constant level. Physical fitness is the ability to perform physical activities without the excessive fatigue of the body [128].

Physical activity and sleep positively correlate with cognitive functions, and especially with executive control and memory consolidation (i.e., the processes that consolidate the acquired information in the brain). Physical activity has a positive effect on the quality of sleep, and especially on its depth, latency, and performance [122,123].

Physical activity is an important element of public health that is used both in the prevention and treatment of various diseases. Regular exercise reduces the risk of cancer, diabetes, and coronary heart disease, as well as the onset of neurodegenerative disorders. The currently available data also indicate a positive effect of exercise and physical activity on sleep quality. Both relatively high-intensity and low-intensity exercise produce significant benefits that are associated with sleep quality [129,130].

Physical activity, and especially regular exercise, can improve the quality of sleep by affecting the adenosine levels and the body temperature; however, when performed too late in the evening, it can cause sleep disruption by increasing physiological arousal. There are also studies that examine the likelihood of sleep improvement through late physical activity that is due to the induced antidepressant, anxiolytic, and body-warming effects [131]. The effect of exercise on the body temperature can be extremely important late in the evening, as there is a decrease in the body temperature when falling asleep, and exercise causes an initial increase in the deep body temperature and it increases the rate of the decrease in the body temperature [129]. The timing of exercises for sleep quality is therefore unclear, as there are many conflicting arguments.

#### 3.3.2. The Effect of Physical Exercise on the Quality of Sleep in People with Insomnia

A study was conducted on 48 patients suffering from insomnia, who were divided into four groups: a control group; a second group, in which people performed moderate-intensity aerobic exercise; a third group, in which high-intensity aerobic exercise was performed; and a fourth group, in which moderate-intensity resistance exercise was performed [129]. In the group in which the participants performed moderate-intensity aerobic exercise, the data from the polysomnogram showed a reduction in the total wakefulness, a delay in falling asleep, and an increase in the efficiency and overall sleep duration. A reduction in anxiety was also observed in those who engaged in moderate-intensity aerobic exercise [129]. The patients with primary insomnia experienced a decrease in their rate of sleep anxiety. Moderate-intensity resistance exercise can cause a reduction in anxiety for up to five hours [132].

Obstructive sleep apnea (OSA) is a condition that occurs during sleep that is characterized by the obstruction of the upper respiratory tract. Obstructive sleep apnea, despite its frequent occurrence, is a clinical condition that is rarely diagnosed [133]. Obstructive sleep apnea is diagnosed by polysomnography, in which the ratio of the total number of apneas and the respiratory shallowness to the total sleep time is assessed [134]. OSA is a growing burden on health care funds, as it is a major source of cardiovascular morbidity and mortality [134]. Patients with obstructive sleep apnea have significantly reduced work performance and quality of life because they often suffer from morning headaches, urination at night, decreased libido, impaired concentration and attention, irritability, depression, sleep fragmentation, neurocognitive disorders, deterioration of the sleep quality, and the occurrence of excessive daytime sleepiness [133].

An easy and cheap method of treating OSA is physical exercise, which alleviates several consequences that are caused by the disease, such as fatigue and cardiovascular disorders. Physical exercise affects weight loss, which also affects the alleviation or resolution of OSA. The mechanisms that attenuate obstructive sleep apnea through physical activity are not yet fully understood, but there are several plausible hypotheses. During physical exertion, the respiratory muscles are stimulated, which leads to structural and metabolic adaptations that increase the resistance to fatigue. It is likely that endurance exercises cause increased activity in the upper respiratory tract, which results in a decrease in the resistance and an increase in the diameter of the upper respiratory tract. Endurance exercises also counteract pharyngeal collapse during sleep [133].

During sleep, fluid accumulates in the neck, which causes an increase in the pressure on the larynx, which can cause the onset of OSA. After aerobic physical exercise, there is a significant reduction in the amount of fluid in the neck [135]. 

People with OSA have reduced durations of slow-wave sleep and greater difficulty achieving slow-wave sleep, as well as increased daytime sleepiness [136]. Physical exercise increases the body temperature, which can make it easier to fall asleep. During physical activity, there is also an increase in the energy expenditure, which affects the extension of the NREM sleep phase [133]. Adult patients (a study of 129 participants) with OSA who practiced physical activity experienced a decrease in daytime sleepiness and increases in the peak oxygen consumption and the sleep performance [137]. 

#### 3.3.3. The Impact of Physical Activity on the Quality of Sleep of Children and Adolescents 

Sleep is a key element for the proper health and development of children. A short sleep duration among preschool children is associated with a higher prevalence of obesity with age [138]. In recent years, the number of obese children has increased sharply. Research conducted by the World Health Organization has shown that about 41 million children under 5 years of age are overweight or obese. It is especially important to maintain a healthy weight in childhood, since overweight and obesity at a young age can adversely affect the mental, physical, and social development of a child. Abnormal body weight in children has a serious impact on the development of diseases in adulthood, such as diabetes, cardiovascular diseases, and cancer [139]. 

International guidelines recommend that infants sleep up to 17 h a day, and that children aged 1–5 years sleep from 10 to 14 h/day [21]. Currently, children sleep less than children did a century ago, and parents report reduced sleep quality observed in their children [138].

A study that analyzed the incidence of insomnia included a group of 700 children aged 5 to 12 years. It was shown that the incidence of insomnia among the children was 19.3%. Insomnia among boys was at a similar level in all the age groups (5–7 years, 8–10 years, and 11–12 years), while, in girls, the 11–12-year age group showed the highest incidence of insomnia symptoms, compared to the 5–7-year and 8–10-year groups. Children with symptoms of insomnia take longer to fall asleep, and they have an increased delay in the REM phase and reduced slow-wave sleep, compared to children without sleep disorders [140].

Higher levels of total physical activity in infants are associated with poorer sleep performance, a shorter total sleep duration, and fewer naps throughout the day. In young children and preschoolers, a higher degree of physical activity has a positive effect on the quality of sleep, as it results in the better quality and the stability of sleep. The intensity of physical activity also affects the quality of sleep. Light physical activity in preschool children is associated with a later bedtime. By contrast, moderate to intense levels of physical activity are associated with a later time and a shorter total duration of sleep. In children aged 1–3 years, higher levels of physical activity are associated with better stability, a shorter overall time to fall asleep, and better sleep quality [138]. 

Playing outdoors, for children aged 1–3 years, is associated with shorter falling-asleep times, fewer wakeups, a shorter total sleeping time, and an earlier bedtime [138]. The physical activity of preschoolers in the open air in the form of play is associated with fewer night awakenings, a shorter time to fall asleep, an earlier sleeping time, and a longer total sleep time. Preschool children who engage in sports have better sleep quality (better sleep performance and earlier time to fall asleep) [138,141].

A study of 91 adolescents (11–19 years old) found that 73.6% of them had trouble maintaining sleep, and that 60.5% had trouble falling asleep [142]. Adherence to physical activity guidelines (60 or more minutes per day of moderate or intense physical activity) resulted in improved quality of sleep and shortened sleep duration [142].

#### 3.3.4. The Effect of Physical Activity on Sleep Quality in Adults

Physical activity and spending time outdoors can be a nonpharmacological means of maintaining proper sleep quality and fighting insomnia [124]. People who practice physical activity have better sleep quality (latency, depth, and sleep performance). Regular physical activity and sleep positively correlate with cognitive functions (executive control and memory consolidation). The available data indicate that women are more likely to engage in nonpharmacological measures to treat insomnia. Furthermore, physically active women had better sleep quality than sedentary women [124].

One study, which involved 305 participants over the age of 40 years, assessed the effects of physical activity on sleep quality. The participants took part in an exercise program that consisted of high-intensity resistance exercise and moderate-intensity aerobic exercise. Pooled analyses of the results showed that physical activity had a positive effect on the sleep quality, which was indicated by a decrease in the PSQI and in the subjective feelings of the participants. An improvement in the quality and latency of the sleep was noted. Those participating in physical activity did not sleep any longer, but they experienced better sleep quality [143,144].

Sleep deprivation is correlated with the occurrence of hypertension, obesity, and stroke. Hypertension is a significant factor in increasing the risk of coronary artery disease, stroke, heart failure, and end-stage renal failure. With an increase in blood pressure above 115/75 mmHg, the risk of cardiovascular disease increases, and it doubles with each increase by 20/10 mmHg. Blood pressure should decrease during the night by 10–20% of the level of the daily blood pressure. In a study that involved 20 hypertensive patients, the participants first performed an exercise test until exhaustion, and then, in random order, they engaged in 30 min of exercise on the treadmill at 7 a.m., 1 p.m., and 7 p.m. This study showed lower blood pressure at night after moderate aerobic exercise at 7 a.m. Physical activity also resulted in improved sleep wellbeing. The deep-sleep phase was increased as a result of the increased energy expenditure during the day, and especially after physical exercise at 7:00 a.m. Therefore, aerobic exercise may be a nonpharmacological method of improving sleep quality [143].

A study was conducted in which a total of 377 women took part [124]. The physical activity was measured by using accelerometers. It was shown that there was a high probability of improving the sleep quality and circadian rhythms through morning exercise. However, the duration of physical activity should be refined, and the clinical significance of the time of day during which physical activity is practiced should be assessed in order to clarify the recommendations for sleep optimization [124].

An important factor that influences the maintenance of proper health in older people is physical activity. Older people are more likely to have mental and physical disabilities and are more likely to have physical limitations compared to younger people. With age, the quality of sleep deteriorates, which is associated with a greater feeling of fatigue during the day, and with reduced comfort and increased mortality [145]. 

The World Health Organization recommends that people practice moderate-intensity aerobic exercise for at least 150 min per week, or high-intensity aerobic exercise for at least 75 min per week [146].

It is assumed that half of the population over the age of 65 years has reduced quality of sleep. A study that surveyed a group of 60 people (22 younger people aged 21–29 years, 16 middle-aged people aged 36–64 years, and 22 older people aged 65–81 years) showed a positive relationship between physical activity and sleep wellbeing in older people [123]. The quality of sleep in the elderly was not related to physical fitness, but to the level of physical activity. Moderate to intense physical activity is particularly associated with improved sleep quality [123]. 

Older people who are physically active are less likely to report symptoms that indicate poor sleep quality. Being more active results in increased sleep efficiency, longer sleep duration, and a reduced delay in falling asleep [147]. 

## 4. Discussion

Studies show that sleep duration has decreased significantly in all age groups [46,47], and sleep disorders and insomnia are diagnosed at all ages [4,9,24,44,45,46]. Currently, many studies suggest that sleep disorders and insomnia increase the risk of cardiovascular disease, obesity, depression, cancer, and infectious diseases [9,10,11,12,50,54].

A reduced sleep time influences poor dietary choices, such as skipping breakfast, eating processed foods that contain fewer vitamins, and eating excessively fatty foods [76], which lead to excessive calorie intake [50]. A major problem is the increase in overweight and obesity in children and adolescents. Studies show that this age group often does not meet the requirements for adequate sleep time [45,46]. The prevention of sleep disorders and insomnia is therefore crucial to prevent noncommunicable diseases, which often start in early childhood. 

Many studies have shown that proper nutrition, physical activity, and fewer stimulants have a positive effect on sleep quality. On the other hand, poor nutrition can lead, in the long term, to inflammation, which is closely associated with insomnia [15]. The nutritional factors that regulate sleep may have different mechanisms of action [35]. Sleep may be affected by individual ingredients (e.g., caffeine), or by a complex of food metabolites. Foods may also influence the commensal microflora, which may lead to the formation of certain bioactive metabolites [41]. Gamma-aminobutyric acid (GABA), which is one of the metabolites of the bacteria, can increase sleep performance and promote sleep [97].

Studies show that a balanced diet has a positive effect on sleep quality [65,66,75]. Foods and meals that contain sufficient protein, carbohydrates, and fats are essential for maintaining the quality of sleep [57,58]. Not only the quantity, but also the quality of the nutrients is important. A sufficient amount of the amino acid tryptophan, which is the precursor of melatonin, has a positive effect on sleep [92]. The scientific evidence points to the role of omega-3 fatty acids, which may positively influence the regulation of serotonin secretion [58]. To improve sleep quality, it is recommended that individuals eat carbohydrate-containing meals with low glycemic indexes, low glycemic loads, and high fiber contents [69,70]. In order to ensure an adequate quality of sleep, processed foods that are high in saturated fatty acids and refined carbohydrates and low in fiber should be avoided [59].

An important factor for sleep quality is its hygiene [101]. Proper sleep hygiene can prevent insomnia and can be a nonmedicinal treatment method. Studies show that the use of stimulants, such as alcohol, nicotine, caffeine, and cannabis, significantly reduces sleep quality [77,84,105,110,119]. 

Physical activity plays an important role in the maintenance of good sleep quality. A sufficient amount of moderate- to high-intensity exercise can improve the quality of sleep and prevent insomnia [126,129]. Physical activity in the late evening, when melatonin is released, may, however, lower melatonin levels [126]. Night exercises, both at high and medium intensities, can even delay the release of melatonin the next evening. However, exercise during the day, regardless of its intensity, does not have a rapid and continuous effect on melatonin secretion [127].

Although there is scientific evidence of an association between disease occurrence and sleep problems, there is little research on the sleep quality in people with noncommunicable diseases in the context of nutrition [68]. 

## 5. Conclusions

Sleep-related issues are a broad and open topic that require further research, and especially because sleep disorders may contribute to the emergence of many chronic diseases. Studies that combine an assessment of the relationship between sleep and diet, physical activity, and the health of the population should be conducted on a wide group of respondents, and especially among people at risk of noncommunicable diseases.

## Figures and Tables

**Figure 1 nutrients-14-01912-f001:**
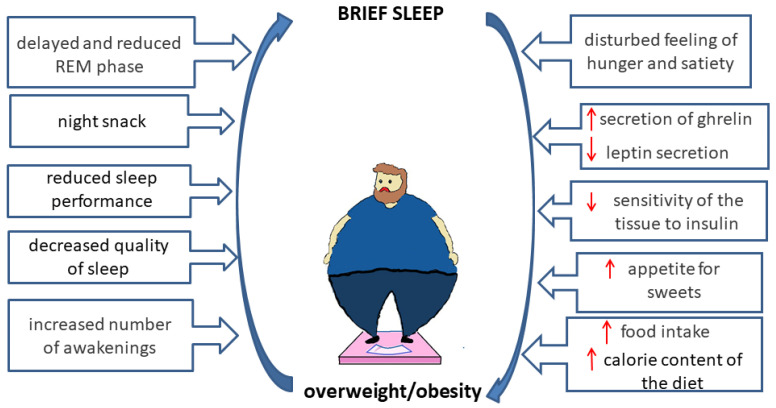
The relationship between inadequate sleep duration and calorie intake.

**Figure 2 nutrients-14-01912-f002:**
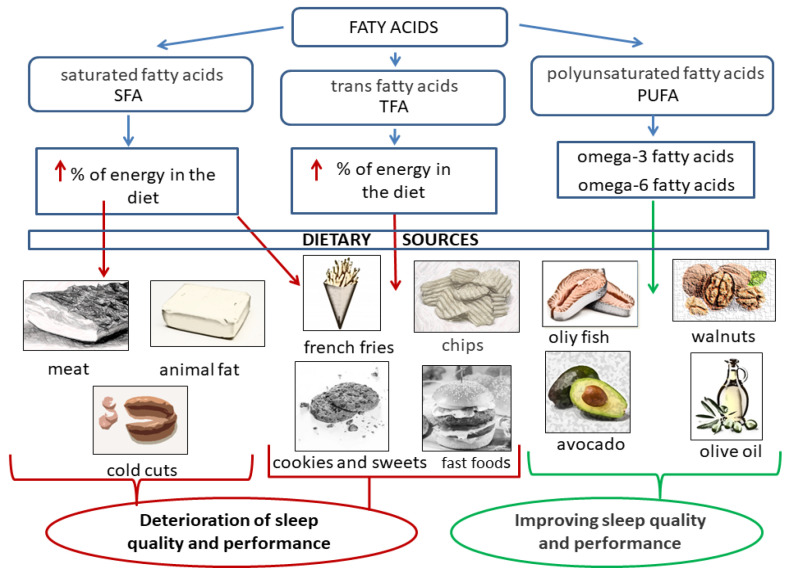
Dietary sources of fatty acids and their association with sleep quality.

**Figure 3 nutrients-14-01912-f003:**
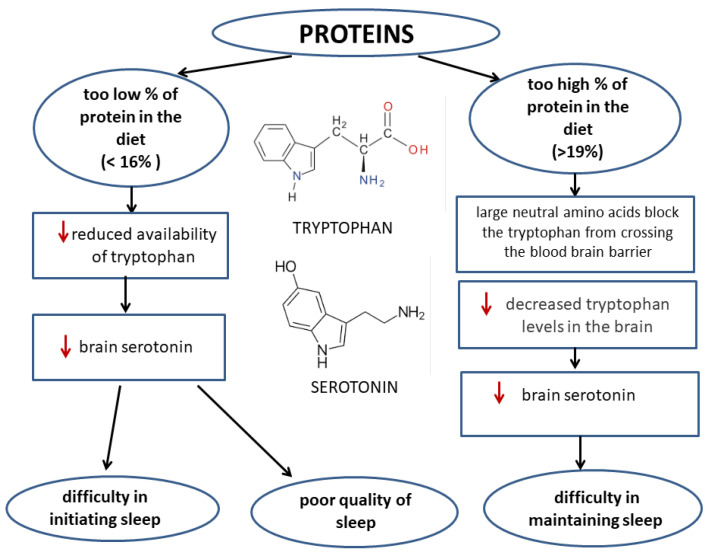
Protein intake and sleep quality.

**Figure 4 nutrients-14-01912-f004:**
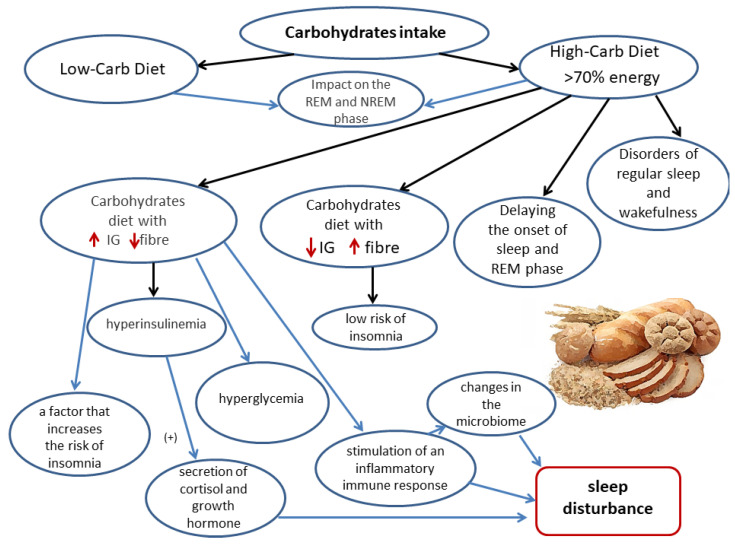
Carbohydrates and sleep quality.

**Figure 5 nutrients-14-01912-f005:**
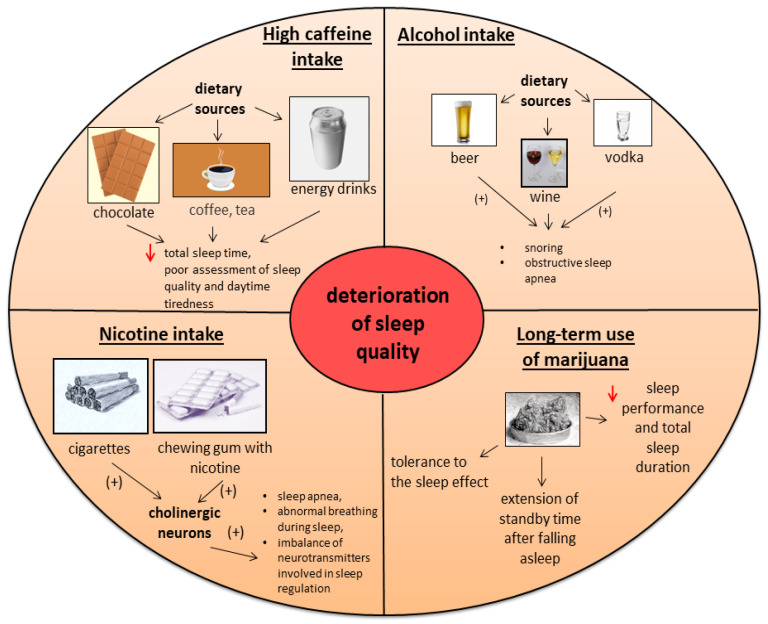
Stimulants and drugs that affect sleep quality.

**Figure 6 nutrients-14-01912-f006:**
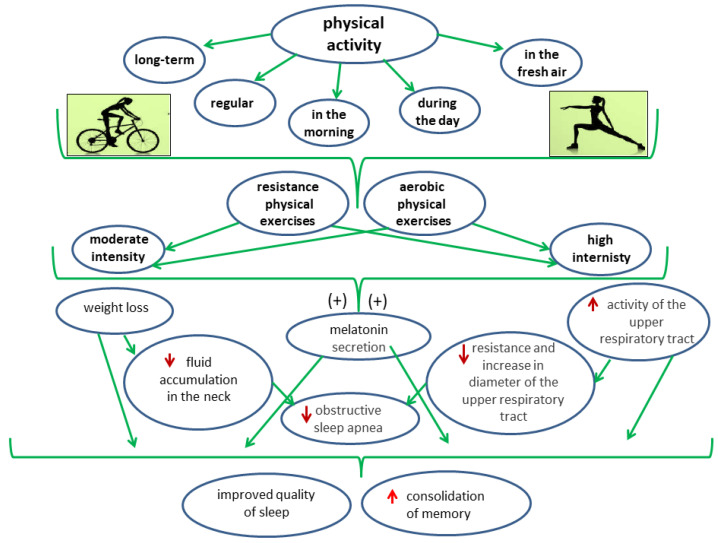
Effects of physical activity on sleep quality.

## Data Availability

Not applicable.

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
