# Peer review of "Sleep Quality: A Narrative Review on Nutrition, Stimulants, and Physical Activity as Important Factors"

_nutrients, 2022, doi:10.3390/nu14091912_

Round 1

Reviewer 1 Report

Dear authors,

The study investigated an issue that affects all ages in our modern society.

The review is well and clear written, however it can be improved as follows:

Line 319 The sentence is unclear. Please re-write the sentence

Line 435 It is unclear. The increased level of caffeine was observed in EEG?? The previous sentence suggests this.

Line 440 Please re-write the sentence. The study" looked at" .. it is unclear.

Line 451 The main source of vitamin D is skin synthesis by exposure to sunlight ( ultraviolet B radiation).  Dietary sources are secondary.  Please specify this.

Line 452  25-hydroxyvitamin D is the main active metabolite of vitamin D, please add, it is unclear

Line 493 It is unclear. The supplements of melatonin have low toxicity?

Line 500  You can find this in yoghurt

Line 501 There are other sources of Gaba than rice. (Briguglio, Matteo et al. “Dietary Neurotransmitters: A Narrative Review on Current Knowledge.” Nutrients vol. 10,5 591. 10 May. 2018, doi:10.3390/nu10050591

Line 528  The sentence must re-written

Line 622 Cannabidiol? What is it? You did not mention

Line 666 " Physical exercise late in the even-666 ing, during an increase in melatonin secretion, can cause a decrease in its concentration." This sentence is unclear. Please re-write.

Line 779  Please reformulate the sentence.

Line 839- 840 Pay attention to the brackets.

Kind regards

Author Response

Thank you very much for your factual and pertinent remarks.
Line 319 The sentence is unclear. Please re-write the sentence Sentence was re-writen.

Line 435 It is unclear. The increased level of caffeine was observed in EEG?? The previous sentence
suggests this. The paragraph was re-written. We are hoping that it is more clear in the current
version.

Line 440 Please re-write the sentence. The study" looked at" .. it is unclear. Sentence was re-written.

Line 451 The main source of vitamin D is skin synthesis by exposure to sunlight ( ultraviolet B
radiation). Dietary sources are secondary. Please specify this. We improved this sentence according
to the Reviewer’s suggestions.

Line 452 25-hydroxyvitamin D is the main active metabolite of vitamin D, please add, it is unclear
The sentence has been improved to clarify the issue.

Line 493 It is unclear. The supplements of melatonin have low toxicity? Supplements of melatonin
has low toxicity. Thank you for this remark.

Line 500 You can find this in yoghurt We added information on food sources of probiotic bacteria.

Line 501 There are other sources of Gaba than rice. (Briguglio, Matteo et al. “Dietary
Neurotransmitters: A Narrative Review on Current Knowledge.” Nutrients vol. 10,5 591. 10 May.
2018, doi:10.3390/nu10050591 Suggested reference has been added as well as GABA sources in the
body text.

Line 528 The sentence must re-written Sentence has been re-written.

Line 622 Cannabidiol? What is it? You did not mention We mentioned that cannabidiol i san active
chemical found in cannabis.

Line 666 " Physical exercise late in the even-666 ing, during an increase in melatonin secretion, can
cause a decrease in its concentration." This sentence is unclear. Please re-write. This sentence has
been improved.

Line 779 Please reformulate the sentence. This sentence has been improved.

Line 839- 840 Pay attention to the brackets. Thank you for this remark. We improved the brackets.

Reviewer 2 Report

Dear authors,

I would like to congratulate you for the attempt to carry out this study, raising relevant questions regarding sleep quality, particularly by discussing the effect of factors such as nutrition, stimulants and physical activity. In my opinion the overall appreciation of the manuscript is ambiguous, considering the importance of the topic, but seriously lacking robustness of the search strategy conducted for considering the inclusion of studies. Also, the manuscript is generally displayed in an inconsistent written English, sometimes I can't understand what it means. An extensive editing is needed as the information is presented in a rather confusing way, and the potential reader often must struggle to understand the text flow. Ideas are not presented in clear and objective form, seriously compromising the text flow. The nature of the review should be clearly stated by the authors. Assuming that the nature of the present study is a narrative review, general guidelines for writing a narrative review were not followed (Baumeister & Leary, 1997; Ferrari, 2015; Siddaway et al., 2019), and no quality assessment of the selected articles was considered (Baethge et al., 2019). Thus, interpretation of the presented results is limited, and generalizations are restricted to the presented references.

Title

Line 2: “Players” should be changed by “factors”.

Abstract

Line 10: “and/or high blood pressure.” Please include and or.

Line 12: “stress anxiety, or use of …”. Please include or.

Introduction

I suggest authors to objectively include all sleep characterization in this section, including sleep phases and duration, insomnia and its risk factors, and functions.

Revies

In my opinion this section presents serious flaws, since that the nature of the review should be clearly stated. Assuming that the nature of the present study is a narrative review, general guidelines for writing a narrative review were not followed (Baumeister & Leary, 1997; Ferrari, 2015; Siddaway et al., 2019), and no quality assessment of the selected articles was considered (Baethge et al., 2019). This option has objective consequences for the studies selection, inclusion and interpretation.

Lines 53 and 57 – Authors refer “studies” when only one study was reported. Please correct.

Line 195 – A reference is needed here.

Lines 478 – 479 – Fragment. Please correct.

Line 563 – “… the next d”. I believe you meant “the next day”. Please correct.

Line 664 – Fragment. Please correct.

Lines 675 – 680 – This definition was originally presented by Caspersen et al. (1985).

Caspersen, C. J., Powell, K. E., & Christenson, G. M. (1985). Physical activity, exercise, and physical fitness: definitions and distinctions for health-related research. Public health reports (Washington, D.C. : 1974), 100(2), 126–131.

Line 804 – Please consider changing the sentence “A study was conducted that completed 20 hypertensive patients”.

Discussion

In the conclusions section authors try to systematize the collected information in the previous sections.

Line 853 – “The improvement of sleep well-being is beneficiary influenced by…”. Fragment. Please correct.

Author Response

Thank you very much for your factual and pertinent remarks.

In my opinion the overall appreciation of the manuscript is ambiguous, considering the importance of
the topic, but seriously lacking robustness of the search strategy conducted for considering the
inclusion of studies. We added search strategy (point 2).

Also, the manuscript is generally displayed in an inconsistent written English, sometimes I can't
understand what it means. An extensive editing is needed as the information is presented in a rather
confusing way, and the potential reader often must struggle to understand the text flow. Ideas are
not presented in clear and objective form, seriously compromising the text flow. We improved
English by the MDPI translating service, but search strategy and discussion, due to limited time for
the revision, were developed when the manuscript was sent to translating service. Nevertheless, we
believe that they are clearly written.

The nature of the review should be clearly stated by the authors. The type of review has been added
in the title and in the search strategy.

Assuming that the nature of the present study is a narrative review, general guidelines for writing a
narrative review were not followed (Baumeister & Leary, 1997; Ferrari, 2015; Siddaway et al., 2019),
and no quality assessment of the selected articles was considered (Baethge et al., 2019). Thus,
interpretation of the presented results is limited, and generalizations are restricted to the presented
references. With great attention we have read the publications about narrative review listed by
Reviewer 2. Most of the recommendations were taken into account in the revision of the manuscript.

Title
Line 2: “Players” should be changed by “factors”. We have made the changes according to the
instructions of the reviewer.

Abstract

Line 10: “and/or high blood pressure.” Please include and or. We corrected it.

Line 12: “stress anxiety, or use of …”. Please include or. We corrected it.

Introduction

I suggest authors to objectively include all sleep characterization in this section, including sleep
phases and duration, insomnia and its risk factors, and functions. We have made the changes
according to the instructions of the reviewer.

Revies

In my opinion this section presents serious flaws, since that the nature of the review should be
clearly stated. Assuming that the nature of the present study is a narrative review, general guidelines
for writing a narrative review were not followed (Baumeister & Leary, 1997; Ferrari, 2015; Siddaway
et al., 2019), and no quality assessment of the selected articles was considered (Baethge et al., 2019).
This option has objective consequences for the studies selection, inclusion and interpretation. We
have considered the reviewer’s suggestions.

Lines 53 and 57 – Authors refer “studies” when only one study was reported. Please correct. We
corrected it.

Line 195 – A reference is needed here. Reference has been added.

Lines 478 – 479 – Fragment. Please correct. We corrected it.

Line 563 – “… the next d”. I believe you meant “the next day”. Please correct. We corrected it.

Line 664 – Fragment. Please correct. We corrected it.

Lines 675 – 680 – This definition was originally presented by Caspersen et al. (1985). The publication
of Caspersen et al. was added in the references.

Line 804 – Please consider changing the sentence “A study was conducted that completed 20
hypertensive patients”. This issue was corrected.

Discussion Discussion was added.

In the conclusions section authors try to systematize the collected information in the previous
sections. A great part of the conclusions has been removed to leave the most important. Some issues
from conclusions were moved to discussion.

Line 853 – “The improvement of sleep well-being is beneficiary influenced by…”. Fragment. Please
correct. This was corrected.

Round 2

Reviewer 2 Report

Dear authors,

I would like to congratulate you for the attempt to incorporate the reviewers’ suggestions and improve the overall quality of the manuscript. The nature of the review was clearly stated by the authors, and general guidelines for writing a narrative review were followed. I particularly appreciated the inclusion of the discussion section. However, I still have some concerns regarding the robustness of the adopted search procedures, which compromises the understanding and generalization of the known results. Nevertheless, a narrative review addresses a topic of interest.

The manuscript is generally displayed in an inconsistent written English, that could still be improved in the final version.

Author Response

Dear Reviewer,

In response to the comment regarding the robustness of the search procedures, we would like to clarify that more than 6000 publications were reviewed in Pubmed by selected keywords (titles and abstracts) before writing the manuscript. We found that about 50 of them met the inclusion criteria. Since usually the review does not assume the search methodology, we did not think it was necessary to include information about the search strategy in the body of the manuscript. At present, it would be difficult to go back with much accuracy and give the exact number of observational studies, experimental studies, and meta-analyses that we were able to review. In any case, we found the most experimental studies. Many of the publications we found were not used in the current review because, after reviewing the content of these papers, it turned out that their topics were somewhat different than the keywords would suggest. Therefore, despite performing an extensive literature review, only some papers were keyed. We would also like to thank you again for pointing out publications that try to systematize the rules of writing review papers. These were very helpful as we revised the manuscript, but we were not able to include all of them.

We thank you for all your suggestions and comments, which allowed us to improve the manuscript and, in the process, to learn a slightly different perspective on the methodology of writing a literature review. We will be able to make full use of them when writing the next manuscript.

As far as the English language of the paper is concerned, we did our best to make the text understandable. However, we are not English-speaking enough to understand certain nuances of the English language. As we mentioned before, the manuscript has been reviewed for the most part by a professional editing service. The remaining part (search methodology and discussion) was reviewed by a native speaker from a local language learning office.